# Recent and Ongoing Research into Metastatic Osteosarcoma Treatments

**DOI:** 10.3390/ijms23073817

**Published:** 2022-03-30

**Authors:** Michael A. Harris, Christine J. Hawkins

**Affiliations:** Department of Biochemistry and Chemistry, La Trobe Institute for Molecular Science, La Trobe University, Melbourne, VIC 3086, Australia; m.harris@latrobe.edu.au

**Keywords:** osteosarcoma, metastasis, metastases, sarcoma, tyrosine kinase inhibitors, immunotherapy, animal models, progression free survival, cancer, metastatic

## Abstract

The survival rate for metastatic osteosarcoma has not improved for several decades, since the introduction and refinement of chemotherapy as a treatment in addition to surgery. Over two thirds of metastatic osteosarcoma patients, many of whom are children or adolescents, fail to exhibit durable responses and succumb to their disease. Concerted efforts have been made to increase survival rates through identification of candidate therapies via animal studies and early phase trials of novel treatments, but unfortunately, this work has produced negligible improvements to the survival rate for metastatic osteosarcoma patients. This review summarizes data from clinical trials of metastatic osteosarcoma therapies as well as pre-clinical studies that report efficacy of novel drugs against metastatic osteosarcoma in vivo. Considerations regarding the design of animal studies and clinical trials to improve survival outcomes for metastatic osteosarcoma patients are also discussed.

## 1. Introduction to Metastatic Osteosarcoma and Its Current Treatment Regimens

Osteosarcoma is the most common primary cancer of the bone. Its emergence is typically driven by mutations in the TP53 and RB1 genes of osteoblastic cells or their precursors in the extremities, such as the tibia or femur [1,2,3]. Osteosarcoma has an annual overall incidence rate of 3.1 per million [4]. It has a bimodal age distribution with a peak incidence of 4.2 per million in children and young adults, decreasing to 1.7 per million in adults aged 25–59, and a subsequent peak of 4.2 per million in older individuals [4]. 

The 5-year survival rate of osteosarcoma was as low as 20% when it was originally defined in the 1950s [5]. The introduction of chemotherapy markedly improved survival. An early randomized trial of adjuvant chemotherapy to treat osteosarcoma explored a combination of doxorubicin, cisplatin, cyclophosphamide, high-dose methotrexate, dactinomycin and bleomycin [6]. The authors reported a 2-year relapse-free survival rate of 66% versus 17% for the control (surveillance) group [6]. Another study later reported comparable survival rates to the multi-drug regimen using only doxorubicin, cisplatin and high-dose methotrexate [7]. A subsequent randomized trial compared the efficacy of doxorubicin and cisplatin versus doxorubicin, cisplatin, cyclophosphamide, high-dose methotrexate, dactinomycin, vincristine and bleomycin [8]. The trial achieved the same survival rate in both treatment groups with improved tolerability in the two-drug regimen group [8]. 

Most osteosarcoma patients currently receive doxorubicin, cisplatin and high-dose methotrexate (referred to as “MAP”) as first-line therapy, alongside limb-sparing surgery to remove the primary tumor. This regimen is associated with a 5-year survival rate of around 70% for localized disease. MAP is also administered as a first-line regimen to patients with metastases at diagnosis, although it is less effective for these patients: between 10% to 40% of them survive for 5 years or more after diagnosis.

Retrospective studies have underscored the utility of metastasectomies for patients with pulmonary metastases, including repeated operations to remove any subsequently detected lung lesions [9,10,11]. Evidence also supports intervention with the less invasive approach of radiofrequency ablation to destroy relatively small, uncalcified pulmonary osteosarcoma metastases [12]. 

A wide variety of systemic therapies have been explored for patients with relapsed or refractory osteosarcoma. Regorafenib (discussed below) is the only drug whose use as a second-line therapy was uniformly recommended by the National Comprehensive Cancer Network on the basis of high level (category 1) evidence, with high-dose ifosfamide plus etoposide and sorafenib receiving recommendations as second-line regimens based on less conclusive evidence [13]. 

Factors that portend particularly poor prognoses for patients with osteosarcoma metastases include: distant bone lesions (compared to the more common lung metastases) [14,15,16,17,18], bilateral (compared to unilateral) pulmonary metastases [16,19] and a greater number of lung nodules [14,20]. Metastases in relapsed patients were reportedly less responsive than metastases within previously untreated individuals, implying that prior chemotherapy exposure (even to different classes of agents) may select for chemo-resistant cells [21].

At diagnosis, around 20% of osteosarcoma patients have detectable metastases. The lungs make up 90% of metastatic sites, with other bones accounting for a further 8% to 10% [22,23]. Many patients who do not have detectable metastases at diagnosis are presumed to have micro-metastases that are undetectable using current imaging techniques [24,25]. One of the challenges in improving the outcome for those with macroscopic metastases is that some phase III clinical trials exclude patients with metastatic disease [23]. This can result in inconsistent treatment across institutes for patients with metastatic disease [23]. 

This review specifically summarizes pre-clinical and clinical trial data relating to novel therapies for metastatic osteosarcoma, to document progress towards improving outcomes for these patients. 

## 2. Results of Published Trials

### 2.1. Chemotherapy

The standard neoadjuvant and adjuvant MAP chemotherapy regimen for patients with localized disease is also administered to patients with metastatic disease, before and after attempting to remove lesions by metastasectomy [23]. 

Various substitutions and additions have been explored, but unfortunately, none have yielded increased survival rates for patients with metastatic osteosarcoma.

Newer generations of anthracycline, such as pirarubicin, that reportedly have better tumor uptake and reduced toxicities have been examined as a substitute for doxorubicin [26]. A trial of 23 patients who were diagnosed with recurrent or refractory osteosarcoma were treated with pirarubicin and cisplatin had a median overall survival rate of 10 months [26]. Although efficacy was limited, the authors recommended follow-up trials with a greater cohort of patients given that all relapsed patients who did respond had metastatic disease [26]. 

Pemetrexed is an inhibitor of thymidylate synthase and has a broader range of action than methotrexate. A phase II trial was conducted to evaluate the efficacy of pemetrexed as a single agent in 32 metastatic osteosarcoma patients [27]. One patient in the cohort had a partial response and the median overall survival rate was only 5.5 months; the study did not meet its primary objective of five responses [27].

Cisplatin’s efficacy is hindered by dose-limiting nephrotoxicity. Trials have evaluated the efficacy of carboplatin, a less toxic platinating agent [28], as a replacement; however, several studies revealed that carboplatin was less effective than cisplatin at treating metastatic osteosarcoma [17,20,29]. Researchers have also attempted to minimize cisplatin-induced toxicity by administering the drug via inhalation to treat pulmonary cancers [30]. A phase I/II trial was established to evaluate the efficacy of inhaled lipid cisplatin in recurrent osteosarcoma patients with pulmonary metastases [31]. Of the 19 patients who received inhaled cisplatin, none exhibited common toxicities associated with cisplatin, such as nephrotoxicity and myelosuppression [31]. Three patients achieved a complete response, one a partial response and seven had stable disease [31]. The best responses were observed in patients with lesions less than 2 cm in size [31]. A subsequent phase II trial (NCT01650090) of inhaled cisplatin for metastatic osteosarcoma with a larger patient cohort was completed in 2018, but no results have been published to date. 

One phase II trial evaluated the addition of ifosfamide to doxorubicin and cisplatin for metastatic disease but found no clinical benefit over standard treatment [32]. Although a total of 30 patients achieved a disease-free status, 21 relapsed with pulmonary metastases an average 15 months later resulting in an overall 5-year survival of only 18% and around 25% in similar study [33,34]. Interestingly, increasing the dose of ifosfamide to treat metastatic disease did not improve patient outcome [35]. 

The use of the topoisomerase II inhibitor etoposide as an additional agent, coupled with MAP plus ifosfamide, has been evaluated in two trials for metastatic osteosarcoma. A phase II/III trial evaluated the efficacy of this combination in 43 patients newly diagnosed with metastatic osteosarcoma [36]. The two-year survival rate was 52% and complete response rate 10% [36]. Significant toxicities were reported in the study, with two patients dying as a result of therapy, 83% of patients developing neutropenia and five suffering sepsis [36]. A later phase I/II trial studied the addition of etoposide to standard chemotherapy plus ifosfamide in 13 patients with metastatic osteosarcoma [37]. Only six patients completed the full regimen, with the remainder requiring dose reduction due to myelosuppression [37]. Median survival for patients with unresectable metastases was 13 months and 31 months for those who underwent successful metastasectomy [37]. Trial data to date suggest that the addition of etoposide into the chemotherapy regimen does not provide a substantial clinical benefit. 

A phase II trial examined the efficacy of the water soluble camptothecin analogue topotecan, after promising data from in vivo testing on osteosarcoma xenografts [38,39]. Twenty-eight newly diagnosed patients with poor prognosis metastatic osteosarcoma were treated with standard chemotherapy plus the topoisomerase inhibitor topotecan, but only one partial response was observed [39]. Topotecan was well tolerated alongside chemotherapy but was not recommended for further study due to its disappointing efficacy [39]. 

A phase II trial examined the efficacy of high-dose thiotpeta (an alkylating agent) with autologous transplantation as an adjuvant to standard chemotherapy in 22 patients with relapsed metastatic osteosarcoma, but reported no significant clinical benefit compared to the 22 control patients who received chemotherapy alone [40]. Gemcitabine is an analogue of deoxycytidine and was trialed as a single agent for advanced sarcoma patients, including two with osteosarcoma, in a phase II trial based on promising activity in animal models of various non-osteosarcoma cancer types [41]. Of the 29 evaluable patients, one leiomyosarcoma patient experienced regression for five months and all other patients had progressive disease [42]. Another trial of gemcitabine documented two cases of progressive disease and four of stable disease in six osteosarcoma patients enrolled in a phase II study for pre-treated sarcoma patients [43]. 

L-alanosine is an inhibitor of de novo adenine synthesis. It was evaluated in 65 carcinoma and sarcoma patients, including seven osteosarcoma patients, three with metastases. Participants’ tumors featured methylthioadenosine phosphorylase deficiency [44], a determinant of sensitivity to L-alanosine [45]. No patients experienced objective responses. Two osteosarcoma patients had stable disease and the other five experienced progressive disease [44].

### 2.2. Bone-Modifying Agents

The bisphosphonate zoledronic acid inhibits bone resorption by suppressing osteoclast differentiation, and also exerts anti-cancer activities through incompletely defined pathways [46]. Zoledronic acid is approved to treat bone metastases from solid tumors. It was first trialed as an additive to chemotherapy to assess toxicity and feasibility in metastatic osteosarcoma patients in 2013 [47]. The authors found that it was safe to administer alongside chemotherapy, but any clinical benefits were difficult to define due to the small number of patients in the trial [47]. A subsequent trial conducted in a similar fashion with 318 patients, including 55 with metastases at diagnosis, found no clinical benefit for patients who received zoledronic acid and chemotherapy versus chemotherapy alone [48].

### 2.3. Stem Cell Rescue

Many osteosarcoma patients initially respond to chemotherapy, but a challenge in maintaining remission is balancing efficacy with the myelosuppressive activity of these treatments [49]. Autologous stem cell rescue combined with high-dose chemotherapy is an alternative treatment protocol for patients unlikely to respond to standard chemotherapy [49]. Several trials have been conducted over the past two decades using stem cell rescue to enable higher doses of treatment to be administered to patients with metastatic osteosarcoma. Unfortunately, survival rates remained unchanged compared to standard treatment protocols, and patients experienced more severe toxicities as a consequence of the increased chemotherapy doses [50,51,52,53].

### 2.4. Immunotherapy

The relatively high level of infiltrating lymphocytes in osteosarcomas compared to other sarcomas [40,54] have made them a promising candidate for immunotherapies [54,55]. One of the earliest trials of sole agent immunotherapy against metastatic osteosarcoma explored the efficacy of inhaled granulocyte macrophage colony stimulating factor (GM-CSF) against recurrent pulmonary metastases [56]. Although the treatment had low toxicity, the authors detected no immunostimulatory effects against pulmonary metastases and no improvement in patient outcome [56].

After encouraging results from treating non-metastatic osteosarcoma patients with the immune modulator liposomal muramyl tripeptide (mifamurtide) [57], addition of this agent to chemotherapy was explored in patients with metastatic disease [58,59]. Mifamurtide activates macrophages and monocytes to stimulate the production of cytokines, which may result in increased anti-tumor activity of infiltrating immune cells [59]. Metastatic osteosarcoma patients treated with chemotherapy plus mifamurtide took significantly longer to relapse than historical controls who just received chemotherapy [60]. However, in the context of a randomized controlled trial, mifamurtide unfortunately did not significantly boost the 5-year survival of metastatic osteosarcoma patients compared to those who received chemotherapy alone [58], although low participant numbers may have precluded detection of a subtle survival benefit [61]. No subsequent trials of mifamurtide in metastatic osteosarcoma have been conducted, but it has been approved by the European Medicines Agency to treat osteosarcoma patients aged between 2 and 30 [62].

The combination of recombinant interleukin 1α and etoposide, which was documented to provoke anti-tumor activity by lymphoid cells [63], was trialed in eight patients with relapsed metastatic osteosarcoma. Two had progressive disease and the rest partial or mixed responses [64]. Although the clinical response was modest, the authors interpreted these results as a good outcome considering the poor prognosis typically experienced by patients who relapse with metastatic disease [64]. Unfortunately, the trial was stopped early due to a halt in the production of recombinant interleukin 1α.

A high proportion of osteosarcomas, particularly pulmonary metastases, express programmed cell death protein-1 ligand (PD-L1). This suggests that metastases may be especially sensitive to PD-1 inhibitors such as pembrolizumab [55,65]. Several trials have evaluated the efficacy of pembrolizumab against metastatic and advanced osteosarcoma, but only one of 49 evaluable patients across three separate trials had a partial response to treatment [66,67,68]. Equally disappointingly, the majority of patients with metastatic disease who participated in trials of PD-1 inhibitors, such as nivolumab, camrelizumab and ipilimumab, experienced progressive disease [69,70,71]. 

Many osteosarcoma patients have human epidermal growth factor receptor 2 (HER2) positive tumors, which formed the basis of a trial evaluating HER2-specific chimeric antigen receptor modified T-cells (CAR T-cells) against HER2-positive sarcomas [72]. Unfortunately, CAR T-cell therapy was no more effective than PD-1 inhibition with 75% of osteosarcoma patients experiencing progressive disease and the remainder only stable disease [72]. Other immunotherapies that rely on the anti-cancer activity of cytotoxic lymphocytes, such as dendritic and T-cell receptor therapies, have failed to improve the outcome of patients with metastatic osteosarcoma [73,74]. 

Although the previously described immunotherapies failed to improve patient outcomes, a prospective study of metastatic osteosarcoma patients who received MAP in addition to IL-2 and lymphokine activated killer (LAK) cell reinfusion yielded more promising results [19]. Of 27 patients who received LAK cell reinfusion and IL-2, 11 remained alive at the time of publication with an overall survival rate of 45% at 130-month median follow up [19].

### 2.5. Drugs Targeting Tyrosine Kinases

Receptor tyrosine kinases represent a broad set of targets, some of which are often overexpressed in osteosarcomas, giving rise to a novel approach of treating metastatic osteosarcoma with tyrosine kinase inhibitors (TKI) [75]. Poor osteosarcoma patient outcome is associated with overexpression of HER2 [76], which prompted evaluation of the HER2 antibody trastuzumab with chemotherapy to treat HER2-positive metastatic osteosarcoma [77]. Patients who received trastuzumab plus chemotherapy had a 30-month overall survival rate of 59%, versus 50% for those who were only treated with chemotherapy [77]. Although trastuzumab was safe to administer to patients receiving chemotherapy, there was no significant difference in patient outcome [77].

OncoLar is a somatostatin analogue that inhibits IGF-1 production, which can suppress the growth of osteosarcoma in vitro [78] and metastasis in vivo [79]. Based on these data, OncoLar was evaluated in a phase I trial of metastatic or recurrent osteosarcoma in 19 patients. It was well tolerated but did not produce a clinical response in any patients [80]. 

Robatumumab is an IGF-1 receptor inhibitor that showed promising pre-clinical activity in subcutaneous patient-derived xenografts [81] and was evaluated in a phase II trial of sarcoma patients with resectable and un-resectable metastases [82]. Of 31 osteosarcoma patients with resectable metastases, one had a complete response and two had a partial response to treatment, whereas none of the 29 osteosarcoma patients with un-resectable metastases responded to treatment [82]. 

Many of the TKIs currently being evaluated have multiple targets, including receptor and intracellular tyrosine kinases that drive pro-tumorigenic pathways, such as angiogenesis and proliferation [83]. Data from clinical trials evaluating TKIs for sarcoma suggest that inhibition of multiple tyrosine kinases is more effective than targeting a single kinase [84]. 

Regorafenib is a multitarget TKI that has been tested for treating metastatic osteosarcoma across two randomized phase II clinical trials. The REGOBONE trial compared regorafenib versus placebo in patients with metastatic osteosarcoma who had failed to respond to chemotherapy. Regorafenib significantly delayed disease progression. Of 26 eligible treated patients, two had partial responses and 15 stable disease with the remainder showing disease progression [85]. SARC024 also trialed regorafenib in a similar fashion and documented a significant improvement in median progression-free survival vs. placebo (3.6 versus 1.7 months) [86]. Surprisingly, despite regorafenib markedly slowing progression in both trials, it unfortunately did not have a significant effect on overall survival [85,86]. As discussed below, imperfect correlations between surrogate endpoints and overall survival raise questions about the optimal primary endpoints for clinical trials. Insights into the basis of such discrepancies may unearth opportunities to build on the anti-progression activity of regorafenib (and perhaps other TKIs) to improve patient survival. 

Sorafenib is another multitarget TKI that has been well studied in osteosarcoma patients. In a phase II trial of 35 patients with metastatic or advanced osteosarcoma, sorafenib treatment led to a reduction in tumor burden in 14% of patients, but only 29% had not progressed within 6 months of commencing treatment [87]. A subsequent phase II trial of sorafenib included everolimus in hope of blocking the mTOR pathway, which may confer resistance to TKI inhibition [88]. Therapy-related toxicity necessitated dose reductions in 66% of patients. Encouragingly, 45% of patients did not suffer disease progression within six months, although only 5% of patients survived for more than two years [88]. 

Cabozantinib is an inhibitor of VEGFR2 and MET and was studied in 43 osteosarcoma patients, of which 39 had pulmonary metastases, in a phase II clinical trial [89]. Similar results to the trials of sorafenib were observed, with a 6-month progression-free survival rate of 33% and only five patients experiencing partial responses [89]. Apatinib is an inhibitor of VEGFR2, which was studied in a phase II trial of 37 patients for advanced osteosarcoma and resulted in a four-month progression free survival rate of 57% [90]. Cediranib, which efficiently targets members of the VEGFR and PDGFR families [91], was assessed in a phase I trial that included four osteosarcoma patients. One experienced a 34% reduction in the size of their pulmonary metastases after two cycles of treatment, but the others did not respond [92]. Similarly, marginal efficacy was observed in the SARC009 phase II trial of dasatinib (a broad-spectrum kinase inhibitor [93]), which included 46 patients with advanced osteosarcoma. Tumors shrank in three of these patients and disease stabilized in another three, and 15% of the osteosarcoma patients survived for at least two years [94]. 

### 2.6. Other Drug Therapies

Other drug classes have been trialed for metastatic osteosarcoma based on promising pre-clinical data, but have yielded generally disappointing outcomes. 

Glembatumumab vedotin is an antibody-drug conjugate that targets an anti-mitotic agent to cells expressing glycoprotein non-metastatic B protein, which is overexpressed in most osteosarcomas [95]. When tested in a phase II trial of 22 recurrent osteosarcoma patients, only one patient had a partial response and one patient’s death may have been attributed to glembatumumab vedotin therapy [96]. Radium 223 dichloride has a specificity for highly mineralized areas [97] and was evaluated in 18 high risk osteosarcoma patients, the majority of whom had bone metastases [98]. Minimal hematological toxicity was observed; however, the 12-month overall survival rate was only 9% [98]. Ecteinascidin 743, which had some activity against drug-resistant osteosarcoma cells in vitro [99], was evaluated in a phase II trial of 23 patients, with 88% bearing pulmonary metastases [100]. As a single agent, its efficacy was limited, with only three patients experiencing minor responses [100]. The mTOR inhibitor ridaforolimus was first evaluated in Ewing sarcoma patients in a phase I trial against advanced solid tumors, which reported 4 partial responses in 32 evaluable patients [101]. These results were the rationale for a phase II trial of ridaforolimus in 212 advanced sarcoma patients including 54 primary bone tumor patients, 51 of whom were diagnosed with pulmonary metastases [102]. Ridaforolimus was more effective in osteosarcoma patients than participants with soft tissue sarcomas. Three osteosarcoma patients achieved partial responses (two confirmed, one unconfirmed) and the bone cancer patients’ 6-month progression-free survival rate was 25% [102]. The authors assayed the levels of eight proteins within the mTOR pathway within tumors, but none predicted clinical responses [102]. 

Despite the numerous trials conducted to evaluate novel therapies for metastatic osteosarcoma, the long-term survival of patients remains unchanged (Table 1). 

## 3. Animal Studies

Many studies have been published evaluating the efficacy of candidate treatments for metastatic osteosarcoma using mouse models. These models are often generated by orthotopic or subcutaneous injection of human or murine osteosarcoma cells into mice, which can metastasize to the lungs over the course of an experiment (Table 2). The presence of a primary tumor in these models makes it difficult to ascertain whether a reduction in metastatic burden is a consequence of a smaller primary tumor in treated mice, seeding fewer cells to the lungs, and/or if the treatment acts directly on established metastases. A reduction in the size of metastases (in addition to their number) in treated mice may imply that the treatment targets osteosarcoma cells within the lungs. Further studies are necessary to confirm these results. Primary tumors are typically removed by surgery in clinical settings, yet this is rarely modeled in animal studies for logistical and ethical reasons.

It is sobering to reflect that the previously described clinical trials failed to recapitulate the exciting efficacy published using pre-clinical animal models. This mismatch may reflect inadequacies of these animal models, in addition to a publication bias favoring reporting of positive outcomes, which presumably would be more prominent in pre-clinical than clinical studies.

Directly seeding osteosarcoma cells to the lungs by intravenous injection can be used to establish a metastatic model without a primary tumor (modeling a context in which a patient’s tumor was surgically removed) [146,147,148,149,150]. These “experimental metastasis” preclinical models of osteosarcoma allow researchers to specifically evaluate whether a novel therapy can effectively target pulmonary metastases. 

Although these models are a powerful tool, it can be difficult to determine if a drug is active against established metastases unless researchers commence treatment after metastases have formed. Authors evaluating therapies such as 3-hydroxyflavone, euxanthone, CXCR4, Timosaponin, TRAIL, Panobinostat, mycophenolic acid, zoledronic acid, rapamycin, parthenolide or quercetin began treatment prior to the injection of cells or shortly afterwards [145,147,148,151,152,153,154,155,156,157,158]. In those studies, efficacy may reflect action against tumor cells within the circulation and/or growing in the lungs. Destroying circulating tumor cells may be desirable, as could preventing cells from acquiring migratory phenotypes, intravasating, extravasating, forming micrometastases in distant sites or activating the proliferation of cells comprising dormant micrometastases. Pathways controlling these key steps of osteosarcoma metastases is an active and fascinating area of research, which has been recently reviewed [159,160]. However, the observation that surgical removal of primary tumors is usually insufficient to cure patients implies that most osteosarcoma patients harbor metastases (subdetectable or overt) when they are diagnosed. As discussed, the prognosis of patients with metastases that are large enough to be detected at diagnosis is much worse than those with ostensibly localized disease. Therapies that suppress early steps in the metastatic process seem unlikely to benefit osteosarcoma patients, particularly those with macro-metastatic disease (“locking the barn door after the horse has bolted”). It is conceivable that identification of molecular features of osteosarcoma cells possessing metastatic activity could facilitate development of targeted therapies capable of specifically eliminating those cells. Improving cure rates for patients with osteosarcoma metastases will probably require agents that can kill osteosarcoma cells comprising established metastases. 

Studies in which authors waited at least one week post-cell injection prior to commencing treatment make a more compelling case for a drug’s efficacy against metastases, as the cells would have presumably arrived at the lungs and begun forming pulmonary tumors. The mTOR inhibitor rapamycin improved the survival rate of xenograft and isograft mice bearing pulmonary metastases compared to those treated with vehicle [161]. The oncolytic adenoviruses Delta24-RGD and VCN-01 both reduced the size and number of pulmonary metastases compared to control-treated mice in models of metastatic osteosarcoma using established cell lines and patient-derived cells [162]. Treatment of mice with the CXCR4 antagonist AMD3100 reduced the number of metastatic nodules in the lungs compared to vehicle-treated mice. In that study, the authors waited four weeks post-cell injection before starting treatment, so it is likely that metastases were established prior to treatment [163]. 

Ideally, researchers would confirm the presence of metastases in mice prior to starting treatment. Very few studies have been published in which the authors detected established metastases in the lungs of mice by microscopy (in pilot experiments) and/or in vivo imaging before beginning treatment. Nasarre et al. observed additive cooperation between PD-1 inhibitor and an antibody targeting Secreted Frizzled-Related Protein 2 (SFRP2) to reduce the number of metastases on the surface of mouse lungs by commencing treatment mice with the antibody in conjunction with a PD-1 inhibitor eight to twelve days after cells were intravenously injected [164]. The proteasome inhibitor ixazomib inhibited the growth of established pulmonary metastases after their presence was confirmed by bioluminescence imaging in two xenograft models, and moderately enhanced survival [165]. The kinase inhibitor sorafenib inhibited the growth of pulmonary metastases and reduced their size in a xenograft model of metastatic osteosarcoma, in addition to inhibiting the growth of subcutaneous osteosarcomas [166]. Sorefenib has since been evaluated to treat metastatic osteosarcoma in two clinical trials, as summarized above [87,88]. Another study found that the Smac mimetic LCL161 targeted established osteosarcoma metastases, inhibiting their growth or inducing their regression or elimination, significantly enhancing survival of mice compared to those treated with vehicle [167].

As an alternative to mouse xenografts and isografts, dogs present a useful model for evaluating new therapies for spontaneous metastatic osteosarcoma. Canine osteosarcoma shares many characteristics with human osteosarcoma, including immune infiltration, microenvironment, genetics and presentation of metastatic disease [168,169]. Although the incidence of osteosarcoma in humans is relatively low, it is 27 times higher dogs, which makes it easier to establish clinical trials with large patient numbers [168]. Some trials have been conducted using dogs as a model for metastatic osteosarcoma to evaluate new therapies, including Auranofin, which improved overall survival combined with standard care [170], and Palladia, which was ineffective as a single agent [171,172]. 

## 4. Current Clinical Trials

Of the clinical trials currently being conducted for metastatic osteosarcoma, almost half are evaluating immunotherapies such as mifamurtide, IL-2 or PD-1 inhibitors (Table 3, Figure 1). Tyrosine kinase inhibitors have also garnered a lot of interest, with multiple trials expanding on the published data of promising agents from this class, including regorafenib [85,86].

## 5. Future Directions

Despite intensive efforts to improve the outcome of patients diagnosed with osteosarcoma, including those with overt metastases, the survival rate for patients reported in clinical trials evaluating new therapies has only modestly increased at best. It seems the long-sought substantial improvements in patient survival will require additional basic biology research and development of drugs engaging relevant molecular targets, followed by well-designed animal experiments and phase II clinical trials, hopefully leading to the identification of promising candidate therapies for evaluation in phase III trials. Optimal clinical trial design will hasten progress towards this goal.

### 5.1. Endpoints and Comparators in Phase II Trials

Mineralization associated with osteosarcomas limits the utility of radiographic imaging for monitoring tumor responses to neo-adjuvant treatment [173,174]. Histological evaluation of resected primary tumors after pre-surgical chemotherapy has strong prognostic value, although the strength of this correlation diminished as treatment intensity increased [175]. Long-term survival after therapy hinges on the elimination of metastases (whether detectable or not), so the predictive power of primary tumor responses to chemotherapy implies that the cells that constitute a patient’s primary tumor and their metastases tend to exhibit similar chemo-sensitivity. It is, therefore, surprising that concordant histological responses between primary tumors and synchronous metastases were only recorded in half of the patients studied by Bacci et al., and heterogenous responses were observed among multiple metastases from some patients [34]. Some drugs appear to exert more toxicity towards subdetectable than overt metastases: replacing cisplatin with carboplatin did not alter the survival rate of patients with localized osteosarcoma, but severely diminished the proportion of patients with metastatic disease who survived for 5 years or more [20]. It may, therefore, be simplistic to assume that novel drugs that can provoke substantial necrosis in primary tumors will necessarily have the same destructive effect on metastases. Likewise, it is conceivable that some classes of drugs may target metastases better than primary tumors. The basis for differential sensitivity between primary osteosarcomas and metastases is unclear, but it could relate to distinct phenotypes of cancer cells that comprise primary versus metastatic tumors, differences in vascularization between primary tumors and metastases (which could influence intratumoral drug concentrations), or variation in the non-cancer cell composition of primary versus metastatic tumors. The strong prognostic influence of the metastatic site (e.g., bone or lymph node versus lung) highlights the differential drug sensitivities of osteosarcomas growing in different anatomical sites. These issues, coupled with the requirement for patients to have measurable tumors, reinforce the disadvantages of using objective response rates as primary endpoints in osteosarcoma trials [176,177]. 

The most meaningful endpoint for clinical trials of osteosarcoma treatments is overall survival (OS). In practice, however, progression-free survival (PFS) (or “event-free survival”; EFS) is more commonly used as a primary endpoint in phase II trials. Concerningly, while PFS and OS data correlated tightly in trials involving Ewing sarcoma patients [178], PFS was a less accurate predictor of OS in trials of localized osteosarcoma patients [179]. As mentioned above, delayed progression may not foretell prolonged survival, “progression” is more subjective than death, so bias must be avoided, and inconsistencies in the timing of progression monitoring between groups may lead to spurious apparent differences in outcomes [180,181]. In trials involving newly diagnosed patients with (seemingly) localized and resectable osteosarcomas, in many of whom “progression” would mean growth of micro-metastases to detectable size, a PFS endpoint may offer the advantage of speed while signifying a prognostically important event. If due caution is paid to its potential pitfalls, on balance PFS may be a reasonable primary endpoint in this context. However, the expected lifespans of patients with recurrent/relapsed disease and/or unresectable osteosarcomas are short. Hence, in trials involving such patients, overall survival may be a more suitable primary endpoint, as it would avoid the risks to data quality associated with PFS without substantially extending the trial’s duration. 

Whether phase II trials should randomize patients to experimental therapies versus placebo is controversial. Some experts contend that single-arm trials with early endpoints can be informative and efficient, as historical benchmarks obviate the need to split relatively scarce patients between two treatment groups. The “New Agents for Osteosarcoma Task Force” devised a structured approach [177] to select agents for evaluation in phase III clinical trials, based on consideration of pre-clinical data and comparison of EFS measures from single-arm phase II trials with historical benchmarks [182]. Using this approach, the task force prioritized assessment of tyrosine kinase inhibitors [177]. However, others doubt the validity of using benchmarks from previous trials [176,183], arguing that participants in phase II trials should be randomized to receive the investigational agent or not, to allow a direct comparison of outcomes. Whether control patients are historical, or randomized participants within the trial, it is crucial to compare treatment efficacy between patients who share similar prognoses. Parameters including prior treatment, tumor resectability, the number of lung metastases and the presence/absence of extrapulmonary metastases could be used to stratify patients in each arm into groups sharing similar prognoses. Using PFS rather than OS as the primary endpoint is more fraught in single-arm than randomized trials. In single-arm trials, the definition of progression (including imaging sensitivity) and monitoring interval, in addition to patient prognoses, must match those used in the historical comparators for outcomes to be directly comparable. By careful selection of comparator datasets from previous trials, and ideally using OS as the primary endpoint, the logistical benefits of single-arm phase II trials may be achievable without unduly compromising the quality of the data and the strengths of conclusions that can be drawn from it.

### 5.2. Predicitive Biomarkers

In addition to providing insight into the potential utility of new therapies, clinical trials that incorporate predictive biomarker assays may reveal avenues for further research that may ultimately improve outcomes for patients. Understanding the physiological, cellular and/or biochemical factors that allow even a small proportion of patients to respond to a particular drug may facilitate personalized treatment approaches based on the features of patients’ primary tumors or metastases. 

Biomarkers were explored to predict patient responses to ridaforolimus, but none of the eight markers assayed varied between patients who did or did not respond to treatment [102]. This may be a consequence of the authors’ use of archived tissue samples or could signify that the complexity of the mTOR pathway makes it difficult to identify a single component responsible for resistance/sensitivity to treatment [102]. 

A study comparing whole genome and exosome sequences between osteosarcoma patients’ primary tumors and their pulmonary metastases revealed that metastases had a much higher tumor mutational burden and genomic instability [184]. Metastases also had a higher level of infiltrating lymphocytes and increased expression of PD-L1, raising hope that they may be more sensitive to immune checkpoint inhibition [185]. In practice, the efficacy of immunotherapy for osteosarcoma has so far been tantalizing but limited. In the phase II trial of apatinib and the anti-PD-1 agent camrelizumab, the authors did not reach their prespecified threshold of 60% 6-month progression free survival [71]. They did, however, note that the two patients with overexpression of PD-L1 experienced durable disease control [71]. Expression of PD-L1 alone as a biomarker to predict patient response to PD-1 inhibitor immunotherapy has so far proved to be unreliable for most cancers [186]. An analysis of all FDA-approved checkpoint inhibitors found that PD-L1 expression only predicted improved responses in 30% of cases across numerous trials [186]. This figure is likely an overestimate as the analysis only examined successful trials that resulted in FDA approval [186]. Counterintuitively, the correlation between overexpression PD-L1 and efficacy of immune checkpoint inhibitors appears to differ between inhibitors that share molecular targets. For example, the PD-1 inhibitor atezolizumab was more effective in improving urothelial carcinoma patient outcomes for those with PD-L1-expressing tumors versus participants with PD-L1-negative tumors [187], whereas pembrolizumab improved patient outcome regardless of PD-L1 status [188]. In some cases, patients experienced durable responses to PD-1 inhibitor therapy, despite having PD-L1 negative tumors [189]. 

Several clinical trials of PD-1 inhibitors for carcinomas suggest that tumor mutational burden (TMB) may be useful for predicting patient response to immunotherapy [190,191,192,193]. High TMB predicted improved patient response to treatment independently of PD-L1 status in non-small-cell lung cancer patients [194]. The implications of high or low TMB in osteosarcoma patients is not well defined; however, osteosarcoma patients with a higher TMB were reported to be more likely to experience longer progression-free survival than patients with a lower TMB [195]. One case study of a patient with pulmonary osteosarcoma metastases and high TMB treated with PD-L1 inhibitor therapy reported a durable response to treatment, with the patient experiencing a remission for at least two years despite discontinuation of treatment due to therapy related toxicities [196]. A separate case study of an osteosarcoma patient with bone and lung metastases and a high TMB experienced a 33-month remission when treated with the PD-1 inhibitor pembrolizumab and controlled disease for 60 months [197]. The results from these two case studies, in addition to robust data from trials with larger cohorts of carcinoma patients, suggest that TMB may be a useful indicator of responsiveness to PD-1 inhibitors in osteosarcoma. Analysis of the TMB of historical samples from patients who were enrolled in PD-1 inhibitor clinical trials would help determine if there was a correlation between the level of TMB and response to treatment.

## 6. Conclusions

The paucity of better-than-expected survival outcomes in osteosarcoma clinical trials summarized above suggests that a dramatic improvement in outcomes for the majority of metastatic osteosarcoma patients will probably require targeting of a novel process or molecule, distinct from those engaged by agents used in clinical trials to date. Hopefully, ongoing pre-clinical research will uncover such game-changing novel targets. Until/unless those approaches bear fruit, extending lifespans for some osteosarcoma patients may hopefully be realized by assembling a panel of therapies that exhibit efficacy in subsets of patients, coupled with development of biomarker assays to enable tailoring of treatments to individual patients.

## Figures and Tables

**Figure 1 ijms-23-03817-f001:**
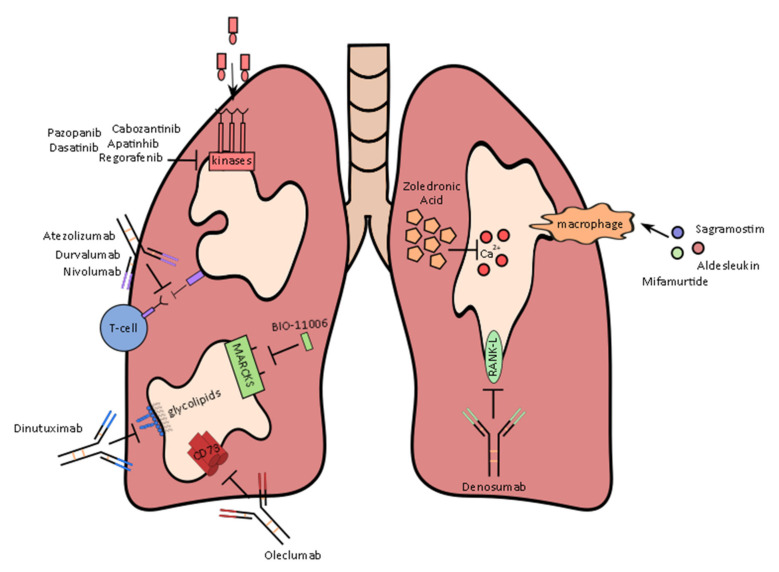
Therapies and their mechanism of action currently being evaluated to treat metastatic osteosarcoma in clinical trials.

**Table 1 ijms-23-03817-t001:** Summary of metastatic osteosarcoma clinical trials and their outcomes. Class of the investigational agent on which the study focused: C, chemotherapy; B, bone modifying agents; S, stem cell rescue; I, immunotherapy; T, tyrosine kinase inhibitors; O, other drug therapies. Responses: CR, complete response; PR, partial response; SD, stable disease; PD, progressive disease. OS, overall survival; PFS, progression free survival; EFS, event free survival.

Therapy	Class	Phase	Outcome	Reference
Pirarubicin-cisplatin	C	N/A	Median OS: 10 months	[26]
Pemetrexed	C	II	Median OS: 5.5 months	[27]
Carboplatin	C	II	3-year OS: 31.9%	[17]
Carboplatin prior to MAP	C	N/A	3-year OS: 65%	[29]
Inhaled cisplatin	C	I	Twelve SD, four PD	[30]
Inhaled cisplatin	C	I/II	Three CR, one PR, seven SD, eight PD	[31]
Cisplatin, ifosfamide and doxorubicin	C	II	Response rate: 33%	[32]
MAP and ifosfamide	C	Follow-up	5-year OS: 24%	[33]
MAP and ifosfamide	C	N/A	2-year OS: 55%	[34]
MAP and ifosfamide	C	N/A	2-year OS: 53%	[35]
MAP, ifosfamide and etoposide	C	II/III	2-year OS: 52%	[36]
MAP, ifosfamide and etoposide,	C	I/II	Median EFS: 13 months	[37]
MAP and topotecan	C	II	5-year OS: 22%	[39]
MAP and high-dose thiotepa	C	II	2-year OS: 66.7%	[40]
Gemcitabine	C	II	Two PD	[42]
Gemcitabine	C	II	four SD, two PD	[43]
L-alanosine	C	II	two SD, five PD	[44]
MAP, ifosfamide, etoposide and zoledronic acid	B	I	2-Year OS: 60%	[47]
MAP, ifosfamide and zoledronic acid	B	III	3-year EFS: 57.1%	[48]
High-dose chemotherapy and stem cell rescue	S	N/A	3-year OS: 20%	[50]
High-dose chemotherapy and stem cell rescue	S	II	Median OS: 34 months	[51]
High-dose chemotherapy and stem cell rescue	S	II	3-year OS: 55%	[53]
GM-CSF	I	N/A	3-year OS: 35.4%	[56]
MAP and mifamurtide	I	III	5-year OS: 53%	[58]
Mifamurtide	I	II	Median OS: 40.5 months	[60]
Interleukin-1α and etoposide	I	II	Three PR, one SD, two PD	[64]
Pembrolizumab	I	II	One PR, six SD, 15 PD	[66]
Pembrolizumab	I	II	Median OS: 5.6 months	[67]
Pembrolizumab	I	II	Median OS: 6.6 months	[68]
Nivolumab with or without ipilimumab	I	II	Without, 1-year OS: 40.4%With, 1-year OS: 54.6%	[69]
Ipilimumab	I	I	One SD, seven PD	[70]
Apatinib and camrelizumab	I	II	6-month PFS: 50.9%	[71]
HER2-CAR-T-cell therapy	I	I/II	Four SD, 13 PD	[72]
Dendritic cell therapy	I	I/II	3-year OS: 2.9%	[73]
CD4^+^ T-cell therapy	I	I/II	One PR	[74]
MAP, ifosfamide, etoposide and trastuzumab	T	II	30-month OS: 59%	[77]
OncoLar	T	I	No clinical responses	[80]
Robatumumab	T	II	Unresectable, median OS: 18 months. Resectable, median OS 8 months	[82]
Regorafenib	T	II	Two PR, fifteen SD, nine PD	[85]
Regorafenib	T	II	Median PFS: 3.6 months	[86]
Sorafenib	T	II	Median OS: 7 months	[87]
Sorafenib and everolimus	T	II	Median PFS: 5 months	[88]
Cabozantinib	T	II	6-month PFS: 33%	[89]
Apatinib	T	II	Four-month PFS: 57%	[90]
Cediranib	T	I	One PR	[92]
Dasatinib	T	II	Four-month PFS: 13%	[94]
Glembatumumab vedotin	O	II	One PR, eighteen PD	[96]
Radium 223 dichloride	O	I	6-month OS: 48%	[98]
Ecteinascidin 743	O	II	Twenty-three PD	[100]
Ridaforolimus	O	II	6-month PFS: 25%	[102]

**Table 2 ijms-23-03817-t002:** Summary of studies reporting the efficacy of novel therapies (not yet evaluated in clinical trials) against metastatic osteosarcoma using pre-clinical models of mice bearing primary osteosarcomas. N, significantly reduced number of individual metastases compared to control mice; S, significantly reduced size of metastases compared to control mice; OT, orthotopic tumor; IM, intramuscular tumor; SC, subcutaneous tumor; I, isograft; X, xenograft.

Drug	Target	Model	Efficacy	Reference
Tegavivint	TBL1	OT, I	N, S	[103]
Anlotinib	Tyrosine kinases	OT, X	N, S	[104]
Exosomal miRNA-206	TRA2B	SC, X	N	[105]
Halofuginone	TGF-β	IM, X	N, S	[106]
Proscillaridin A	JAK2/STAT3	SC, X	N	[107]
TH1579	MTH1	IM, X	N	[108]
Infliximab	TNFα	OT, X	N	[109]
Squalenoyl-Gemcitabine and Edelfosine	Pyrimidine analogue, cell membrane	OT, X	N	[110]
LCL161	cIAP1/2	IM, X	N, S	[111]
Gefitinib	EGFR	OT, I	N, S	[112]
miRNA-1225	YWHAZ	SC, X	N	[113]
Hybrid liposomes	Cell membrane	SC, I	S	[114]
Anti-midkine antibody	Midkine	IM, X	S	[115]
XGFR antibody	IGF-IR and EGFR	OT, X	N	[116]
Melittin	Wnt/ β-catenin	OT, X	N	[117]
PR619	Ubiquitin specific proteases	OT, X	N	[118]
Auranofin	Thioredoxin system	SC, X	S	[119]
Oleanolic acid	PI3K/Akt/mTOR/NF-κB	OT, X	S	[120]
Anginex gene therapy and radiation	Endothelial cell growth	OT, X	S	[121]
PEDF	Angiogenesis	OT, X	S	[122]
Decitabine	Estrogen Receptor Alpha	OT, X	S	[123]
CD47 blockade	CD47	SC, I	S	[124]
Gemcitabine, rapamycin	Cytidine analogue, mTOR	SC, I, OT, X	N	[125]
Lycorine	Wnt/β-catenin	OT, X	N, S	[126]
BMTP-11	IL-11:IL-11Rα	OT, X	N, S	[127]
Esculetin, fraxetin	M2 macrophage differentiation	SC, I	N	[128]
Anti-Tim3, PD-L1, anti-OX-86 and surgery	Tim3, PD-L1, anti-OX-86	SC, I	N, S	[129]
AEG-1 siRNA	Astrocyte elevated gene-1 RNA	SC, X	S	[130]
L-MTP-PE, zoledronic acid	Nonspecific immunomodulation, hydroxyapatite	OT, X, I	N	[131]
Eribulin	Microtubules	SC, I	S	[132]
Apatinib	VEGFR2	SC, X	N	[133]
Xanthoangelol	Stat 3 phosphorylation	SC, I	N	[134]
Edelfosine nanoparticle	Phosphatidylinositol phospholipase C	OT, X	N, S	[135]
LB100	phosphatase 2A	SC, X	N	[136]
Disulfiram	Aldehyde dehydrogenase	OT, I	N, S	[137]
Nivolumab	PD-1	SC, X	N, S	[138]
Zoledronic acid	hydroxyapatite	SC, I	S	[139]
Midostaurin	Tyrosine kinases	OT, I	N	[140]
Meloxicam	COX-2	SC, I	N	[141]
Pigment epithelium-derived factor	Angiogenesis	OT, X	N	[142]
PEDF derived peptides	Angiogenesis	OT, X	N	[143]
VEGF-SiRNA	VEGF	SC, X	N	[144]
Parthenolide	NF-κB	SC, I	S	[145]

**Table 3 ijms-23-03817-t003:** Summary of ongoing metastatic osteosarcoma clinical trials.

Clinical Trials.Gov Identifier	Therapy	Phase	Patients Enrolled	Completion Date
NCT01590069	Aerosolized Aldesleukin	I	70	December 2022
NCT01953900	iC9-GD2-CAR-VZV-CTLs	I	26	October 2034
NCT02517918	Metronomic chemotherapy (cyclophosphamide and methotrexate and zoledronic acid)	I	26	March 2022
NCT03612466	CycloSam^®^ and external beam radiotherapy	I	20	September 2024
NCT04877587	Gemcitabine and ascorbate	I	20	May 2023
NCT00788125	Dasatinib, Ifosfamide, Carboplatin, and Etoposide	I/II	143	December 2021
NCT03811886	Natalizumab	I/II	20	October 2023
NCT02243605	Cabozantinib S-malate	II	90	June 2019
NCT02357810	Pazopanib Hydrochloride and Topotecan Hydrochloride	II	178	June 2022
NCT02389244	Regorafenib	II	132	March 2023
NCT02470091	Denosumab	II	56	September 2022
NCT02484443	Dinutuximab and Sargramostim	II	41	March 2020
NCT03063983	Metronomic chemotherapy (cyclophosphamide and methotrexate)	II	158	January 2022
NCT03643133	Mifamurtide and chemotherapy	II	126	October 2028
NCT03742193	Apatinib and Gemcitabine-docetaxel chemotherapy	II	43	September 2022
NCT04183062	BIO-11006 and Gemtax	II	10	November 2023
NCT04668300	Oleclumab and Durvalumab	II	75	June 2024
NCT04690231	Apatinib, etoposide and ifosfamide	II	79	June 2021
NCT04803877	Regorafenib and Nivolumab	II	48	June 2026
NCT05019703	Atezolizumab and Cabozantinib	II	40	December 2027
NCT03932071	Zoledronic Acid	IV	150	January 2023

## Data Availability

Not applicable.

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
