# Peer review of "Recent and Ongoing Research into Metastatic Osteosarcoma Treatments"

_ijms, 2022, doi:10.3390/ijms23073817_

Round 1
Reviewer 1 Report
I would think this manuscript is an excellent comprehensive review of preclinical to clinical research on metastatic osteosarcoma. A few of the concerns I have noted are as follows.
1. It would be better to have a Table summarizing the results of published trials. That table would be very useful for readers to refer to.
2. In the Introduction, there is a mention of 0.8 - 10 in 100,000. Does this mean an incidence rate for the child to adolescent population? If so, 10 in 100,000 seems too high in frequency, but is the data correct?
3. I would think it may be an exaggeration to say that osteosarcoma in the elderly is usually due to Paget's disease of bone. Please reconsider.
Author Response
We thank the reviewer for their thoughtful feedback on our manuscript. Below we have itemized the changes we have made to address their comments.
1. It would be better to have a Table summarizing the results of published trials. That table would be very useful for readers to refer to.
- Thank you for the suggestion. We have included the requested table: it’s Table 1 in the revised manuscript.
2. In the Introduction, there is a mention of 0.8 - 10 in 100,000. Does this mean an incidence rate for the child to adolescent population? If so, 10 in 100,000 seems too high in frequency, but is the data correct?
- Published incidence rates vary. We have modified this text to mention the epidemiology data from a large study published by Mirabello et al (lines 23-27).
3. I would think it may be an exaggeration to say that osteosarcoma in the elderly is usually due to Paget's disease of bone. Please reconsider.
- We recognise this was an overstatement, so have removed the reference to Paget’s disease.
Reviewer 2 Report
The review described the pre- and clinical trials in patients with metastatic osteosarcoma. I have some comments on this review. I recommend the authors add the present strategies for metastatic osteosarcoma.
1 Introduction
How do the authors think the indication of metastasectomy or interventional procedure (e.g. RFA)? I think the patients who could undergo it may prolong their survival.
2.1 chemotherapy
The authors should add the present strategies for metastatic osteosarcoma. Which is 1st line? 2nd line? I know the outcome is not satisfactory. But we know the present outcome because we have only administered chemo-drug now.
2.5 Drug targeting to TK
Please make a Table for further understanding the outcome using them.
Author Response
We thank the reviewer for their thoughtful feedback on our manuscript. Below we have itemized the changes we have made to address their comments.
The review described the pre- and clinical trials in patients with metastatic osteosarcoma. I have some comments on this review. I recommend the authors add the present strategies for metastatic osteosarcoma.
- We have expanded our summary of current treatments for metastatic osteosarcoma in the first section (lines 40-58) have altered the first subheading to reflect this focus(line 20).
1 Introduction: How do the authors think the indication of metastasectomy or interventional procedure (e.g. RFA)? I think the patients who could undergo it may prolong their survival.
- The revised manuscript highlights the value of metasectomies and radiofrequency ablation (lines 45-49).
2.1 chemotherapy: The authors should add the present strategies for metastatic osteosarcoma. Which is 1st line? 2nd line? I know the outcome is not satisfactory. But we know the present outcome because we have only administered chemo-drug now.
- The revised manuscript includes more explicit informaiton regarding first-line versus second-line treatments (line 40-44 and 50-55).
2.5 Drug targeting to TK: Please make a Table for further understanding the outcome using them.
- Tyrosine kinase inhibitors have been included in the new table (Table 1).